# Deconvolution of single-cell multi-omics layers reveals regulatory heterogeneity

Longqi Liu[1,2,3], Chuanyu Liu[1,2,4], Andrés Quintero [5,6], Liang Wu[1,2,4], Yue Yuan[1,2,4], Mingyue Wang[1,2,4], Mengnan Cheng[1,2,4], Lizhi Leng[7,8], Liqin Xu[1,2], Guoyi Dong[1,2], Rui Li[1,2,3], Yang Liu[1,2,4], Xiaoyu Wei[1,2,4], Jiangshan Xu[1,2,4], Xiaowei Chen[2], Haorong Lu[2], Dongsheng Chen[1,2], Quanlei Wang[1,2,4], Qing Zhou[1,2], Xinxin Lin[1,2], Guibo Li [1,2], Shiping Liu [1,2], Qi Wang[5], Hongru Wang[9], J. Lynn Fink[1], Zhengliang Gao[10], Xin Liu [1,2], Yong Hou [1,2], Shida Zhu[1,2], Huanming Yang[1,11], Yunming Ye[3], Ge Lin[7,8,12], Fang Chen[1,2,13], Carl Herrmann[5,6], Roland Eils [6,14], Zhouchun Shang [1,2,10] & Xun Xu[1,2,15]

Integrative analysis of multi-omics layers at single cell level is critical for accurate dissection of cell-to-cell variation within certain cell populations. Here we report scCAT-seq, a technique for simultaneously assaying chromatin accessibility and the transcriptome within the same single cell. We show that the combined single cell signatures enable accurate construction of regulatory relationships between *cis*-regulatory elements and the target genes at single-cell resolution, providing a new dimension of features that helps direct discovery of regulatory patterns specific to distinct cell identities. Moreover, we generate the first single cell integrated map of chromatin accessibility and transcriptome in early embryos and demonstrate the robustness of scCAT-seq in the precise dissection of master transcription factors in cells of distinct states. The ability to obtain these two layers of omics data will help provide more accurate definitions of "single cell state" and enable the deconvolution of regulatory heterogeneity from complex cell populations.

[1] BGI-Shenzhen, Shenzhen 518083, China. [2] China National GeneBank, BGI-Shenzhen, Shenzhen 518120, China. [3] Harbin Institute of Technology Shenzhen Graduate School, Xili University Town, Shenzhen 518055, China. [4] BGI Education Center, University of Chinese Academy of Sciences, Shenzhen 518083, China. [5] Division of Theoretical Bioinformatics, German Cancer Research Center (DKFZ), Heidelberg 69120, Germany. [6] Health Data Science Unit, Heidelberg University Hospital, Heidelberg 69120, Germany. [7] Institute of Reproductive & Stem Cell Engineering, Central South University, Changsha 410078, China. [8] Key Laboratory of Stem Cells and Reproductive Engineering, Ministry of Health, Changsha 410078, China. [9] Institute of Vertebrate Paleontology and Paleoanthropology, Chinese Academy of Sciences, Beijing 100044, China. [10] Department of Regenerative Medicine, Tongji University School of Medicine, Shanghai 200092, China. [11] James D. Watson Institute of Genome Sciences, Hangzhou 310013, China. [12] National Engineering and Research Center of Human Stem Cell, Changsha 410078, China. [13] Laboratory of Genomics and Molecular Biomedicine, Department of Biology, University of Copenhagen, 2100 Copenhagen, Denmark. [14] Center for Digital Health, Berlin Institute of Health and Charité, Berlin 10117, Germany. [15] Institute for Stem cell and Regeneration, Chinese Academy of Sciences, Beijing 100101, China. These authors contributed equally: Longqi Liu, Chuanyu Liu, Andrés Quintero, Liang Wu, Yue Yuan. Correspondence and requests for materials should be addressed to R.E. (email: roland.eils@bihealth.de) or to Z.S. (email: shangzhouchun@genomics.cn) or to X.X. (email: xuxun@genomics.cn)

The rapid proliferation of single-cell sequencing technologies has greatly improved our understanding of heterogeneity in terms of genetic, epigenetic, and transcriptional regulation within cell populations[1]. We, and others, have developed single-cell whole genome[2], exome[3,4], methylome[5], and transcriptome[6,7] technologies and applied these approaches to analyzing the complexity of cell populations in tumorigenesis, developmental process, and cellular reprogramming[8]. Meanwhile, single-cell epigenome techniques, including single-cell ChIP-seq[9], ATAC-seq[10,11], DNase-seq[12], and Hi-C[13,14], have been developed to decipher histone modifications, transcription factor (TF) accessibility landscapes, and 3D chromatin contacts, respectively, in single cells. These techniques provide important information on regulatory heterogeneity by assessing chromatin structure across various cell types.

Measuring the epigenomic and transcriptomic characteristics of single cells is important for understanding the maintenance and conversion of cell fates, as well as manipulating cell fates into different lineages[15]. The regulation of these processes involves sequential events including the binding of TFs to *cis*-regulatory elements (CREs) and the recruitment of chromatin regulators, resulting in changes of chromatin structure and activation or repression of cell-type-specific genes[15]. Single-cell ATAC-seq and RNA-seq represent a great opportunity to study how TFs and epigenomic features induce transcriptional outcomes that influence cell fate determinations. For example, combined analyses of datasets by these two approaches have enabled characterization of subtypes in mouse tissues[16] or during human hematopoietic differentiation[17]. However, it still remains challenging to integrate the two approaches experimentally in individual cells, thus hampering a full understanding of regulatory association between these two layers. Here, we present scCAT-seq (single-cell chromatin accessibility and transcriptome sequencing), a technique that integrates single-cell ATAC-seq and RNA-seq to measure chromatin accessibility (CA) and gene expression (GE) simultaneously in single cells. scCAT-seq employs a mild lysis approach and a physical dissociation strategy to separate the nucleus and cytoplasm of each single cell. Thereafter, the supernatant cytoplasm component is subjected to the Smart-seq2 method as described previously[7]. The precipitated nucleus is then subjected to a Tn5 transposase-based and carrier DNA-mediated protocol to amplify the fragments within accessible regions (Fig. 1a). Beyond parallel CA and GE profiling in the same single cell, scCAT-seq will be particularly useful for analyzing samples when the amount of input material is limited.

## Results

**Simultaneous profiling of accessible chromatin and gene expression in single cells.** We applied scCAT-seq to the K562 chronic myelogenous leukemia cell line, which has been widely used in the ENCODE project. We sorted single-cell and multi-cell samples (e.g., 500 cells) into wells of 96-well plates using flow cytometry. Empty wells were used as negative control. Samples were then processed using the scCAT-seq protocol. qPCR analysis confirmed the successful capture of single-cell nuclei during library preparation (Supplementary Figure 1a). We generated combined CA and GE profiles from a total of 192 samples. Of the 176 single-cell profiles, 74 (42.0%) of them passed both CA and GE data quality control criteria (Supplementary Figure 1b and Methods).

For scCAT-seq-generated CA data, we obtained an average of $2.1 \times 10^5$ uniquely mapped, usable fragments from single cells (Supplementary Data 1 and Supplementary Figure 1c, d). Similar to bulk ATAC-seq[18], the CA fragments showed fragment-size periodicity corresponding to integer multiples of nucleosomes

(Supplementary Figure 1e) and are strongly enriched on accessible regions (Fig. 1b and Supplementary Data 1). We found that about 9% of the fragments were mapped to the mitochondrial genome (Supplementary Figure 1f), which is largely reduced in comparison with standard bulk ATAC-seq studies (typically over 30%)[18]. Pearson correlation analyses revealed our single-cell profiles could reproduce features of bulk profiles (Supplementary Figure 1g). In comparison with the published scATAC-seq profiles by Buenrostro et al.[10], we obtained a higher number of usable fragments per single cell but with lower signal-to-noise ratio (Supplementary Figure 1h). However, the correlation between single cells increases remarkably (Supplementary Figure 1h), suggesting that scCAT-seq is able to capture the chromatin features more accurately.

For mRNA data generated by scCAT-seq, we obtained an average of 4.6 million reads covering over 8000 genes (GENCODE v19, TPM > 1), which is comparable with published scRNA-seq profiles by Pollen et al.[19] (Supplementary Figure 1j and Supplementary Data 1). Consistent with published Smart-seq profiles, our mRNA data showed full coverage of the transcript body (Fig. 1b), enabling identification of transcript isoforms and not merely gene expression quantification. The aggregate profile was close to the RNA-seq profile obtained from 500 cells (Pearson correlation value > 0.9, Supplementary Figure 1i), suggesting that scCAT-seq is able to accurately quantify GE of single cells. The density of CA and GE reads of all single cells surrounding a constitutively accessible region showed that scCAT-seq data could recapitulate major features obtained by separately performed bulk ATAC-seq and RNA-seq (Fig. 1c).

GE regulation is associated with the structure of the CREs (e.g., histone modifications, DNA methylation) and the binding of *trans*-factors (e.g., TFs, epigenetic modifiers)[20]. Therefore, we examined the overall distribution of single-cell CA fragments across different genomic contexts, as well as the expression levels of the putative regulated genes. We observed that the CA fragments were enriched at CREs with active histone modifications (e.g., H3K27ac, H3K9ac, and H3K4me3), whereas repressive or inaccessible regions (e.g., H3K27me3 and H3K36me3-associated regions) showed lower fragment density (Fig. 1d). We also observed other association patterns between CA and GE. For example, we found low levels of CA fragments on H3K36me3-associated regions but high levels of GE fragments. This is not surprising because H3K36me3 is known to be enriched on the active gene body which is occupied by nucleosomes and rendered inaccessible[20]. Notably, genes with bivalent marks (co-enrichment of H3K4me3 or H3K4me1 and H3K27me3) showed similar level of accessibility as active genes (co-enrichment of H3K4me3 or H3K4me1 and H3K27ac, but lack of H3K27me3), and both of them showed higher levels of accessibility than inactive genes (enrichment of H3K27me3, but not H3K27ac, H3K4me1, and H3K4me3). Conversely, the expression levels of bivalent genes were remarkably lower than active genes and were similar to those of inactive genes. We also investigated the distribution of CA fragments across genomic contexts bound by different TFs and found an overall consistent pattern between CA and GE level. Notably, we observed substantial decrease of expression levels of genes associated with binding of EZH2 while the accessibility level showed just a moderate change (Fig. 1e). This pattern is similar to that of bivalent genes and is consistent with the role of EZH2 which, as part of the repressive polycomb complex, catalyzes H3K27me3. Thus, the combined signatures from scCAT-seq well reflect known processes and are useful to assess the transcriptional state of genes within different genomic contexts. This approach is undoubtedly of high value for many biological applications, for example, studying the heterogeneous transition of bivalent genes during development or cellular reprogramming.

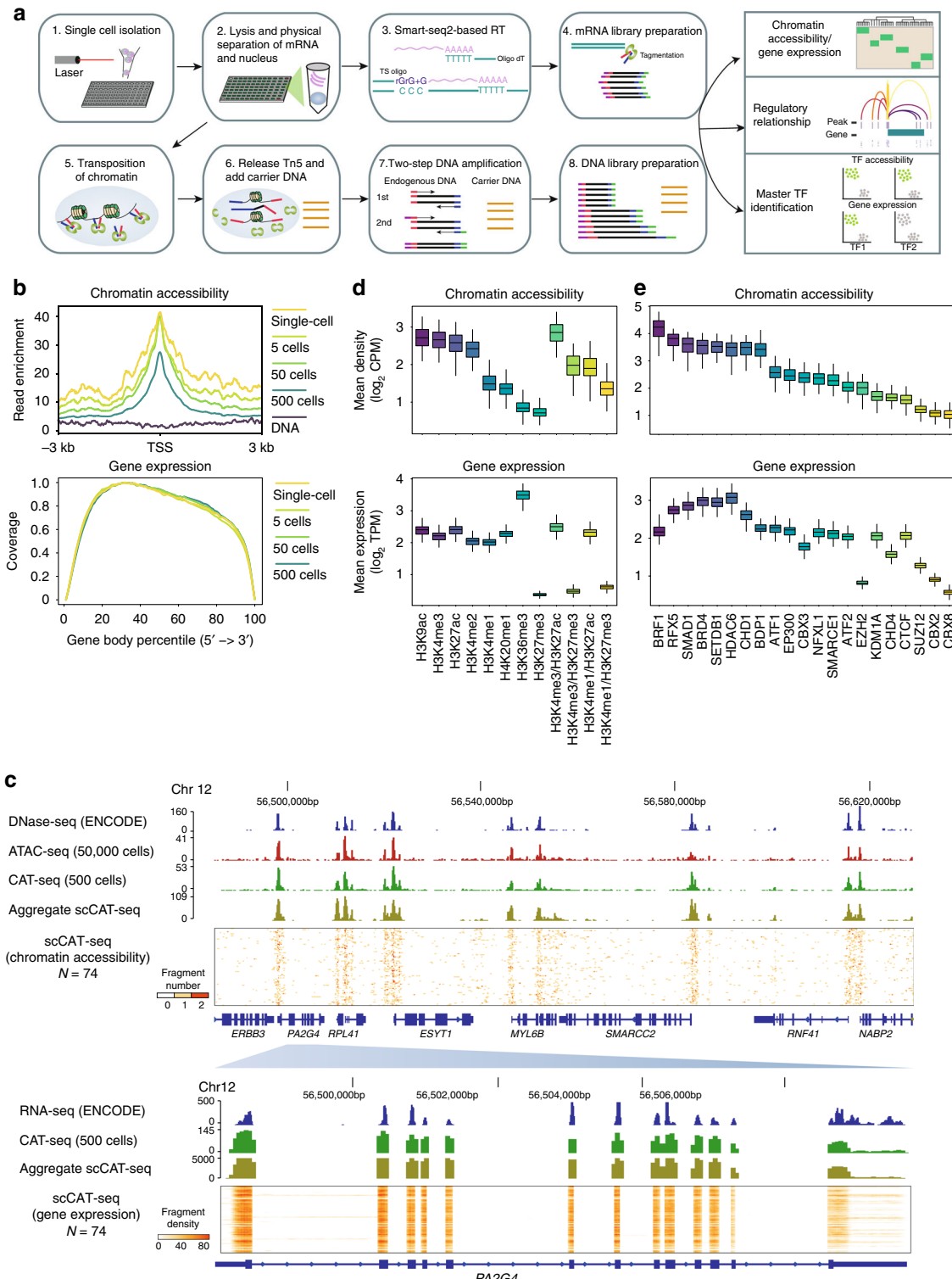

**Fig. 1** scCAT-seq provides an accurate genome-wide measure of both chromatin accessibility and gene expression. **a** Overview of the scCAT-seq protocol. **b** Top panel: chromatin accessibility read enrichment around the transcription start site (TSS). Bottom panel: coverage of mRNA reads along the body of transcripts. Titration series (one single-cell, 5 cells, 50 cells, 500 cells) were marked by the indicated colors. All profiles were generated using the scCAT-seq protocol with the indicated number of cells as input. **c** A representative region showing a consistent pattern of chromatin accessibility and gene expression across datasets generated using different number of input cells. The bulk ATAC-seq track was generated using 50,000 K562 cells. The DNase-seq and bulk RNA-seq data of K562 cells were downloaded from ENCODE. The scCAT-seq tracks are chromatin accessibility (upper) and gene expression read density (bottom) from a total of 74 K562 single cells. **d** Top panel: mean chromatin accessibility read density around regions that are enriched by the indicated individual or combined histone modifications. Bottom panel: mean expression level of genes associated with regions that are enriched by the indicated individual or combined histone modifications. **e** Top panel: mean chromatin accessibility read density within regions that are bound by the indicated transcription factors. Bottom panel: mean expression level of genes associated with regions that are bound by the indicated transcription factors

We further validated our approach by generating different batches of scCAT-seq profiles from two additional ENCODE cell lines: HeLa-S3 cervix adenocarcinoma and HCT116 colorectal carcinoma cell lines (Supplementary Data 1). To test the feasibility of scCAT-seq in real tissue samples, we also generated profiles from two lung cancer patient-derived xenograft (PDX) models (Supplementary Data 1). One is derived from a moderately differentiated squamous cell carcinoma patient (PDX1) and the other one from a large-cell lung carcinoma patient (PDX2). Principal components analysis (PCA) on both CA and GE profiles resulted in separation of cells from different origin (Supplementary Figure 2a, b). A comparison of our datasets with published profiles revealed that the differences across protocols and batches had a substantially smaller effect than difference across cell types (Supplementary Figure 2c, d).

**Establishment of regulatory relationships between CREs and genes in single cells.** Next, we explored the dynamic associations between the two omics layers across single cells. We first tested the correlation between accessibility level of single CREs and their expression of the putative target genes in each of the three cell lines, and the hypothetical cell population merged from them. As expected, we identified remarkably more positive correlations (Pearson correlation > 0; FDR < 10%) than negative correlations (Supplementary Figure 3a), which is consistent with the known relationship between CA and GE in bulk profiles[21].

An earlier study showed the co-variability of accessibility between CREs across single cells defines regulatory domains highly concordant with observed chromosome compartments, which provides an alternative approach to the discovery of regulatory links[10]. However, it still remains impossible to directly infer the transcriptional outcomes of each chromatin accessible region. Given the overall positive correlation between CA and GE, we reasoned that the co-variability between accessibility of individual elements and expression of genes could enhance discovery of regulatory links that influence transcription. To this end, while employing the reported strategy using scATAC-seq[10] (strategy 1, Fig. 2a), we proposed two additional strategies for inferring regulatory relationships (strategies 2 and 3, Fig. 2a). For strategies 1 and 2, regulatory relationships between chromatin accessible regions and target genes were identified based on scATAC-seq and scCAT-seq data, respectively. Based on the scATAC-seq data, regulatory relationships for every gene were assigned when the Spearman correlation of the accessibility of CREs located at the promoter and distal peaks was above 0.25 (strategy 1, Fig. 2a and Methods). Likewise, for the scCAT-seq data, the regulatory links were assigned if the Spearman correlation between the GE and the accessibility of distal CREs was above 0.25 (strategy 2, Fig. 2a and Methods). However, these regulatory relationships were defined across all cells. In order to more accurately depict the regulatory relationship between chromatin and genes, in strategy 3, single-cell-specific regulatory relationships between genes and their nearby accessible regions were assigned using the scCAT-seq data as follows: (i) identification of active TFs for every cell by SCENIC[22] using the normalized GE matrix; (ii) identification of active accessible regions by matching the binding motifs of active TFs to accessible chromatin regions; and (iii) assignment of regulatory relationships after applying a Wilcoxon test to determine if the presence of a nearby active accessible region was associated with a significant change in the target GE ($P$-value < 0.05) (Fig. 2a and Methods).

By applying the 3 strategies to single cells of the 3 cell lines, we found that strategy 3 identified the largest number of regulatory relationships (62,769), compared to strategy 1 (46,813) and strategy 2 (21,219) (Fig. 2b). Over 1/3 of the regulatory relationships from scATAC-seq based method (strategy 1) were shared by those from scCAT-seq based method (strategies 2 and 3), suggesting strong synergistic effects between regulation at chromatin and transcriptome levels. Nevertheless, although a similar correlation approach was used in strategies 1 and 2, strategy 2 identified a lower number of regulatory relationships, suggesting a possible decoupling between accessibility at the promoter and the expression of the gene. Notably, we also observed a large fraction of regulatory relationships specifically identified by each method, which suggests that different information can be obtained from single-omics and combined analysis.

To assess the accuracy of the regulatory links inferred by each method, we next counted the regulatory relationships that could be verified by chromatin interaction analysis by paired-end tag sequencing (ChIA-PET)[23]. Encouragingly, using the ChIA-PET interactions of the three widely used cell types (K562, HeLa-S3, and HCT116)[24], we observed higher proportion of validations in scCAT-seq based method (strategies 2 and 3) than that in scATAC-seq based method (strategy 1) in all three cell types (Fig. 2c). These suggest that the co-variability between CA and GE layers could better reflect higher-order chromatin structure than co-variability between CREs. One explanation is that regulatory relationships inferred from scATAC-seq may result from either chromatin interactions or from co-binding of master TFs without interaction, while those inferred from scCAT-seq could be considered to be "functional" regulatory relationships as including information from both chromatin interactions and co-binding of master TFs. Therefore, based on the largest number of validated regulatory relationships, strategy 3 outperformed the other strategies (hereafter, the "regulatory relationship" indicates those identified only by strategy 3). The distribution of distance between each pair of peak and gene in all regulatory relationships showed higher enrichment in proximal regions than distal regions (Supplementary Figure 3b), suggesting that GE tends to be regulated by proximal elements which is consistent with earlier findings[25].

To assess whether the regulatory relationships in each single cell reflect cell type-specific features, we generated a binary matrix where columns represent single cells and rows represent all identified regulatory relationships between accessible sites and genes, and the entries indicate the on or off state of each regulatory relationship in each cell. We applied a non-negative matrix factorization (NMF) method, implemented in the R package Bratwurst[26], to decompose the matrix into different signatures that could distinguish single-cell identities. As expected, NMF clustering of the regulatory relationships identified signatures containing numerous cell type-specific regulatory relationships, resulting in clear separation of the three cell types (Fig. 2d, e, and Supplementary Figure 3c). For example, *SAMSN1* is a known oncogene, preferentially expressed in the blood cancer, multiple myeloma[27]. We observed highly specific regulatory relationships around *SAMSN1* in K562, a myelogenous leukemia cell line (Fig. 2e), revealing a strong association between its expression and accessibility of CREs. This observation again reconfirmed the importance of epigenetic mechanisms during progression of tumors. Likewise, we generated regulatory relationship matrix for single cells from PDX tissues and clustering of the matrix clearly separated these two type of cells (Fig. 2f, g, and Supplementary Figure 3d). Interestingly, we also observed a subpopulation of cells showing specific regulatory relationships in PDX2 (Fig. 2f, g), likely reflecting the regulatory heterogeneity present in real tissues.

**Integrated single-cell epigenome and transcriptome maps of human pre-implantation embryos.** We next explored the

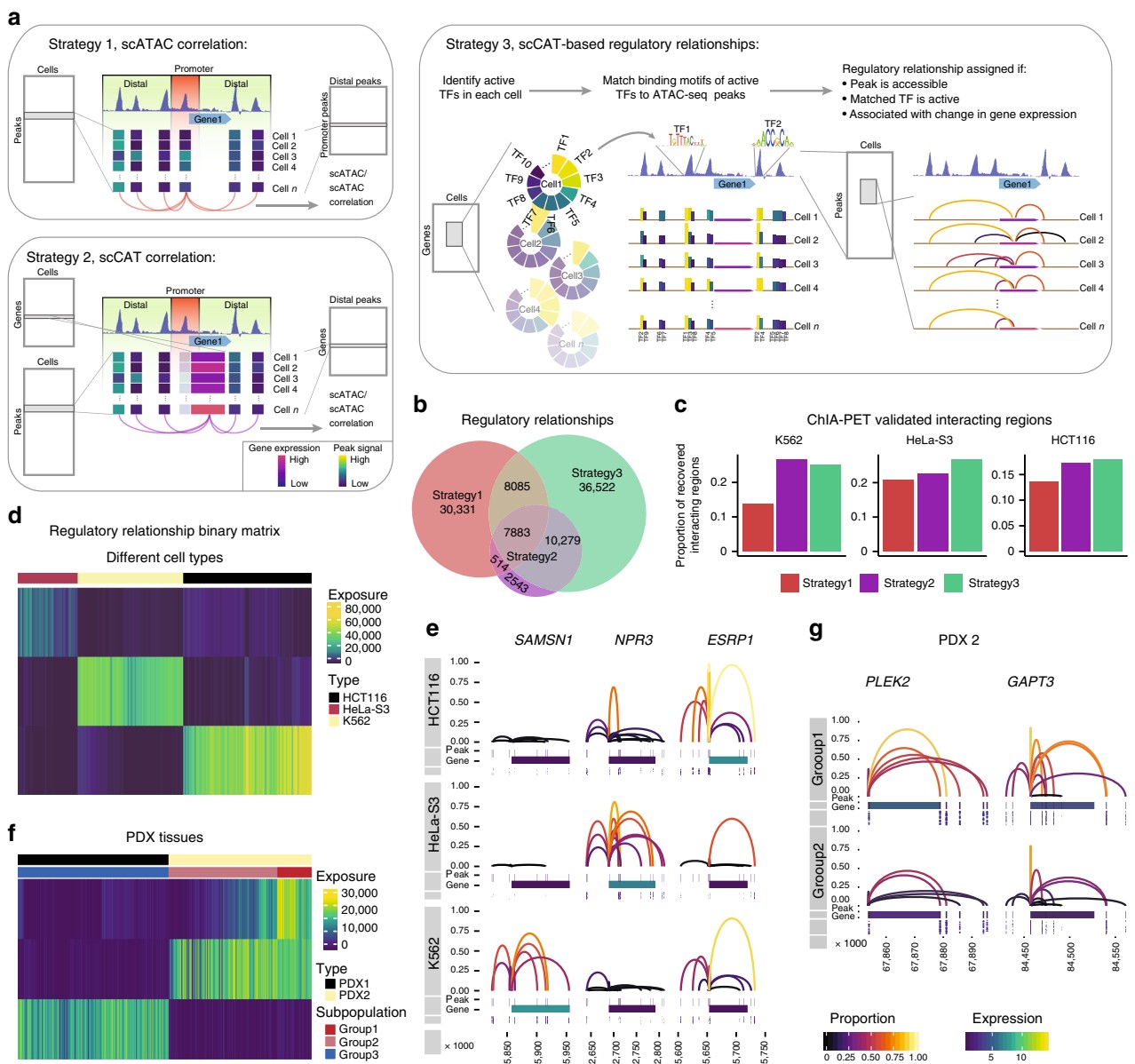

**Fig. 2** Inferring regulatory relationships between CREs and genes by scCAT-seq. **a** Overview of three strategies for inferring regulatory relationships. Strategy 1: regulatory links for every gene were assigned when the Spearman correlation of the signal of peaks located at the promoter and distal peaks was above 0.25. Strategy 2: the regulatory links were assigned if the Spearman correlation between the gene expression and the signal of distal peaks was above 0.25. Strategy 3: active transcription factors for every cell were identified by SCENIC, then active regions were identified by matching the binding motifs of active transcription factors to accessible regions. Then regulatory relationships were assigned after applying a Wilcoxon test to determine if the presence of a nearby active accessible region was associated with a significant change in the target gene expression (*P*-value < 0.05). **b** Venn plot showing the number of overlapping regulatory relationships identified by the three strategies. **c** Proportion of ChIA-PET validated regulatory relationships identified by the three strategies in K562 (left), HeLa-S3 (middle), and HCT116 (right) single cells. **d**, **f** Heatmaps showing exposure scores of all cells to each signature identified by the NMF clustering of regulatory relationship binary matrices of cell lines (**d**) and PDXs (**f**). The exposure score represents the contributions of the signatures to the different samples. **e**, **g** Regulatory relationships for the indicated genes in single-cell groups of the cell lines (**e**) and PDX2 (**g**). Each panel contains three tracks: the top track shows the regulatory relationship between one peak and the gene (linking them with an arch), where the height and color of the arch show the proportion of cells that share the regulatory relationships; the middle track shows the genomic location of the gene and the associated peaks, where the color of the gene shows the mean expression in each cell type; the bottom track shows the accessible states (on and off) for each peak in each single cell

potential of scCAT-seq in the characterization of single-cell identities in continuous developmental processes. The human pre-implantation embryo development is a fascinating time that involves dramatic changes in both chromatin state and transcriptional activity. However, it has only been investigated at either the chromatin or the RNA level due to the lack of truly integrative approaches[28]. By using clinically discarded human

embryos (Methods), we generated scCAT-seq profiles for a total of 110 individual cells, and successfully obtained 29 quality-filtered profiles from the morula stage and 43 from the blastocyst stage (success rate 65.5%) (Fig. 3a, Supplementary Figure 4a and Supplementary Data 1). To explore the regulation relevant to each stage, we identified ~100 K regulatory relationships and generated a matrix of regulatory relationships across all single

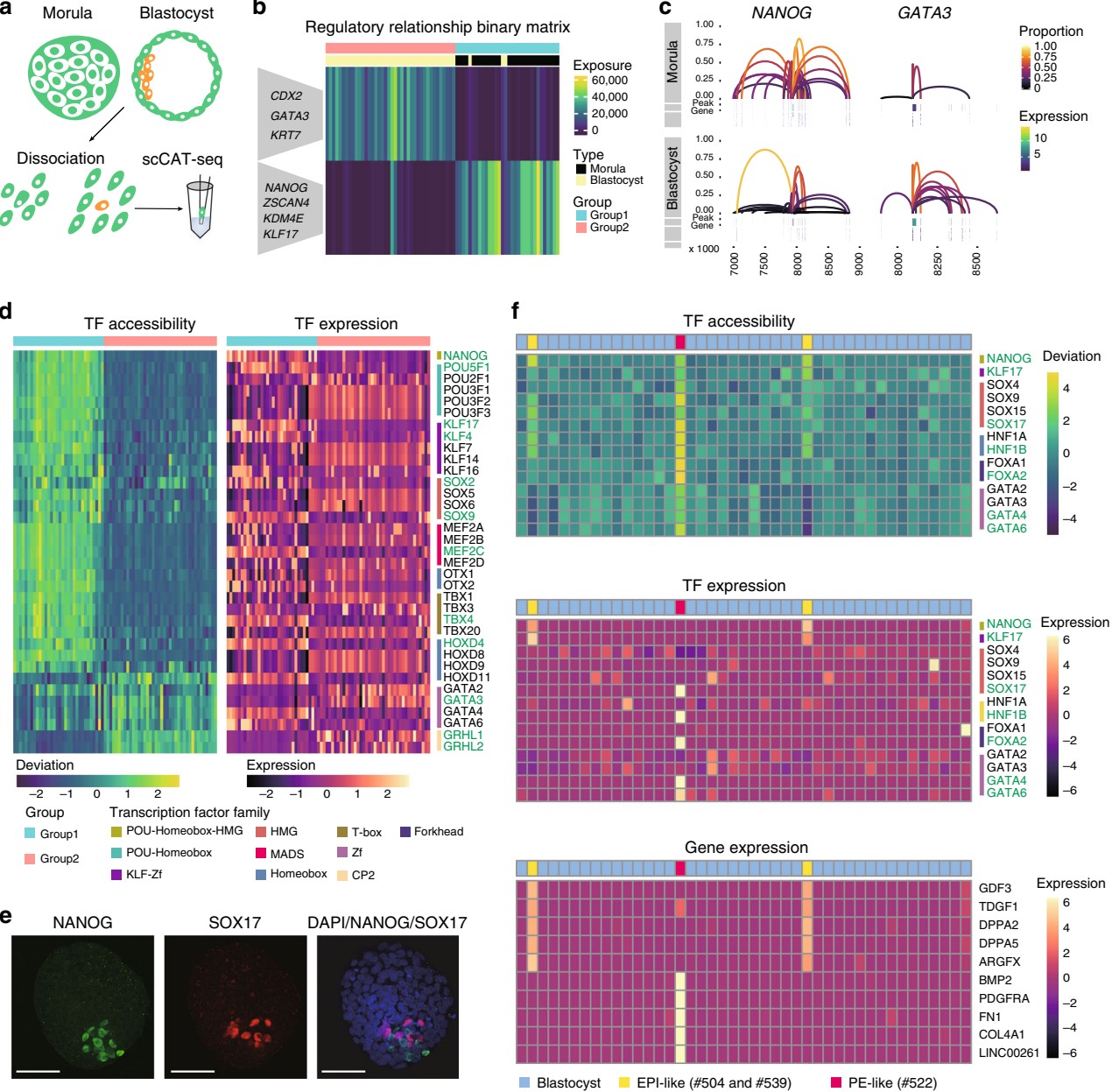

**Fig. 3** scCAT-seq enables precise characterization of single-cell identities in human pre-implantation embryos. **a** A workflow showing the generation of scCAT-seq profiles of human pre-implantation embryos. **b** Heatmap showing exposure scores of all cells to each signature identified by the NMF clustering of regulatory relationship binary matrix of human embryos. Example genes are shown. **c** Regulatory relationships for the indicated genes in single cells of the morula and blastocyst stage. **d** Heatmaps showing accessibility deviation (left) and expression level (right) of the indicated TFs. The TFs colored in green were the ones showing consistent patterns in accessibility and gene expression. **e** Immunofluorescence imaging of the human blastocyst stage embryo using the indicated antibodies (left to right: NANOG, SOX17 and merged DAPI/NANOG/SOX17). Scale bar represents 50 μm. **f** Top and middle panels: Heatmaps showing the accessibility deviation (top) and expression level (middle) of the indicated TFs in single cells of blastocyst-stage embryos. Bottom panel: heatmap showing the expression level of the indicated genes. The TFs coloured in green were the ones showing consistent patterns in accessibility and gene expression

cells as described above. NMF clustering analysis of the matrix showed separation of all single cells into two main groups (groups 1 and 2), corresponding to these two stages (Fig. 3b). The heatmap of exposure scores to each signature revealed activation of regulatory relationships of pluripotency markers (such as NANOG and KLF17) in the morula, and trophectoderm (TE) markers (such as CDX2 and GATA3) in the blastocyst stage[28] (Fig. 3b, c and Supplementary Figure 4b, c), which strongly suggests that the expression of these markers is activated/maintained by epigenomic states[28].

The transition between cell fates largely depends on TFs, which bind to CREs and recruit chromatin modifiers to reconfigure chromatin structure[15]. Single-cell chromatin accessibility data provide a great opportunity to find the key TFs in individual cells[10,17]. However, TFs of the same family often share similar motifs, which makes it difficult to determine the key TFs of functional specificity. Previous efforts have proposed computational algorithms to integrate CA and GE data, but the accuracy remains uncertain because the analyses are based on separate multi-omics datasets[16,17].

We reasoned that functionally relevant master TFs in each cell type should be determined by integrated omics data obtained by scCAT-seq. We applied chromVAR[29], a method for inferring TF accessibility with single-cell CA data, to compute the deviations of known TFs across all single cells. This method identified TF motifs with high variances (Supplementary Figure 4d), dividing all single cells into two main groups (Supplementary Figure 4e), in agreement with the clustering results on regulatory relationships (Fig. 3b). We observed that motifs from the POU-Homebox, SOX-HMG, and KLF-zf families showed high deviation scores in cells of the group 1, while motifs from GATA-zf and GRHL-CP2 families showed high deviation scores in cells of the group 2 (Fig. 3d). To determine the master TF from each family, we next integrated the expression level of these TFs. Interestingly, we found that the well-known pluripotency factors (such as NANOG, POU5F1, SOX2, KLF4, and TBX4), as well as early markers (such as KLF17), both showed relatively high levels of CA and GE in cells of the group 1, whereas other TFs of the same families (such as POU3F1, SOX5, KLF7, and TBX1) showed opposite trends (Fig. 3d). These results are highly consistent with the features of the pluripotent morula cells, which are the main component of group 1. We also found GATA3, but not GATA4 and GATA6, to show a specific role in the group 2, which contains cells from the blastocyst stage. This is in agreement with the important role of GATA3 during differentiation of trophoblast[30]. In addition, we also observed similar results from other TFs of the same families, such as SOX9, HOXD4, MEF2C, and GRHL1, suggesting they likely playing critical roles in these two groups (Fig. 3d). Overall, these results suggest that our integrated method could increase the power of discovery of functionally relevant TFs at single-cell resolution.

The blastocyst stage consists of inner cell mass (ICM) and TE lineages. During the maturation of blastocysts, the ICM segregates into pluripotent epiblast (EPI) and primitive endoderm (PE) cells[31]. The number and size of ICM cells vary across blastocysts, and are important for the grading of embryos that determine the success of implantation[32]. Notably, the clustering of both regulatory relationships and TF accessibility deviation showed that 3 (#504, #539, #522) out of the 43 blastocyst cells are similar to morula cells (Fig. 3b). This reveals the pluripotency feature of these three single cells in the blastocyst stage and suggests that they might be from ICM cells (hereafter termed ICM-like cells). This result is also supported by our data based on immunostaining in a human blastocyst embryo, which showed a comparable small proportion using the known, lineage-specific markers NANOG (EPI) and SOX17 (PE) (Fig. 3e).

We next sought to validate the ICM-like cells by molecular features based on their two omics signatures. It is known that OCT4 is initially expressed in all cells within the ICM, and becomes restricted to the EPI in the late blastocyst[31]. Interestingly, although OCT4 is not a general marker of the blastocyst stage (Fig. 3d), it has a higher deviation score in the three single cells compared with other cells in the blastocyst (Supplementary Figure 4f). Notably, two of them (#504 and #539) showed even higher deviations from the other single cell (#522) (Supplementary Figure 4f), which may describe the segregation into EPI (#504 and #539) and PE (#522) lineages (hereafter termed "EPI-like" and "PE-like" cells).

We next attempted to support this hypothesis by identifying the key TFs in the EPI- or PE-like cells. Encouragingly, in addition to enrichment of OCT4, we also observed specific enrichment of the well-known EPI-specific regulators, such as NANOG, and KLF17, in EPI-like cells (Fig. 3f), while the PE-like cell showed high activity of the well-known PE regulators, such as SOX17, HNF1B, and FOXA2 (Fig. 3f). The other members of the same families (such as SOX9, FOXA1, and HNF1A) are not likely

to be the key regulators because of the inconsistent patterns of CA and GE. Further supporting this conclusion, the well-known non-TF markers were also found to be highly specific to each cell type, including GDF3, TGDF1, DPPA2, DPPA5, and ARGFX in EPI-like cells and BMP2, PDGFRA, FN1, COL4A1, and LINC00261 in PE-like cells[33] (Fig. 3f). Although the EPI- and PE-like cells are similar to morula cells, the above markers tend to be transcriptionally active in EPI- or PE-like cells based on CA and GE profiles (Supplementary Figure 4g), suggesting distinct pluripotent states in the morula and blastocyst stages. Taken together, these results indicate that our integrated approach can faithfully identify the two distinct subtypes from the same origin. The robustness of scCAT-seq in the precise definition of single-cell identities would be particularly useful for characterization of cells that are rare within complex cell populations.

## Discussion

In summary, our work demonstrates that scCAT-seq is able to provide high resolution epigenomic and transcriptomic portraits of individual cells. We showed that the accessibility levels of both regulatory elements and particular TFs are positively correlated with the GE program. This provides a highly relevant insight into regulatory relationships, one which is not possible based on individual omics profiles. We proposed a method to establish regulatory relationships by linking CREs to the putative target genes, resulting in a larger numbers of high-confidence regulatory interactions compared with state-of-the-art methods. The cell-specific regulatory relationship is a new feature that enables the direct discovery of gene centered 3D regulatory patterns in certain cell populations, thus providing the basis for a more comprehensive study of regulatory mechanisms at the single-cell level. Moreover, we generated the first integrated single-cell epigenomic and transcriptomic maps during pre-implantation embryo development. The robustness of scCAT-seq in the characterization of distinct cell states reveals the great potential of scCAT-seq in faithful identification of new cell types in complex cell populations, which enables a better understanding of developmental abnormalities caused by either genomic variants or environmental influences. Overall, we show that scCAT-seq is a highly promising tool for the joint study of multimodal data of single cells, paving the way to a thorough assessment of regulatory heterogeneity in a variety of clinical applications, including pre-implantation screening.

## Methods

**Cell culture**. K562 chronic myelogenous leukemia cells (ATCC) were cultured in RPMI-1640 medium (Gibco) supplemented with 1x penicillin–streptomycin (Pen-Strep, Invitrogen) and 15% fetal bovine serum (FBS, Gibco). HCT116 colorectal carcinoma cells (ATCC) were cultured in Iscove's Modified Dulbecco's Medium (Gibco) supplemented with 1x Pen-Strep and 15% FBS. HeLa-S3 cervix adeno-carcinoma cells (ATCC) were cultured in medium containing Dulbecco's Modified Eagle Medium (Gibco) supplemented with 1x Pen-Strep and 15% FBS.

**Bulk ATAC-seq library preparation**. Bulk ATAC-seq libraries were generated using a modified protocol based on previous study[18]. Briefly, 50,000 cells were collected and washed with cold 1x PBS. Cells were centrifuged and resuspended using 50 μl of ice-cold lysis buffer (10 mM Tris-HCl, pH 7.5, 10 mM NaCl, 3 mM MgCl$_2$, and 0.1% IGEPAL CA-630 (Sigma)). Then the lysate was centrifuged and resuspended in 50 μl of transposition reaction mix (10 μl 5 X TAG buffer (50 mM TAPS-NaOH, pH 8.5, 25 mM MgCl$_2$, 50% DMF), 1.5 μl in-house Tn5 transposase (0.8 U/μl) and nuclease-free water (NF-water)), and incubated for 30 min at 37 °C. The subsequent steps of were performed as previously described[18].

**Single-cell isolation from patient-derived xenograft**. The human lung cancer patient-derived xenograft (PDX) models were bought from Shanghai LIDE Biotech Co., Ltd. with written informed consent and institutional approval. The PDX samples used in this study were approved by the Institutional Review Board (IRB) on Human Subject Research and Ethics Committee in the Shanghai LIDE Biotech Co., Ltd., China. One of the PDX models is derived from a moderately

differentiated squamous cell carcinoma patient and the other one from a large-cell lung carcinoma patient. In brief, 50–90 mg PDX tumor pieces were implanted subcutaneously on the right flank of each mouse. The tumor tissues were isolated when the mean tumor size reached ~400 mm³ and then enzymatic digested to single-cell suspension for FACS sorting.

**Collection of human pre-implantation embryos**. All embryos were obtained from the donors undergoing in vitro fertilization (IVF) treatments at in compliance with the Ethics Committee of Reproductive & Genetic Hospital of CITIC-XIANGYA using standard clinical protocols as described previously[34]. All volunteers signed an informed consent document.

The morula- and blastocyst-stage embryos were produced by conventional intracytoplasmic sperm injection (ICSI) of these donated oocytes by donated sperm from the same couple. Embryos were transferred to the wells of pre-equilibrated EmbryoSlide (Vitrolife, Sweden) and cultured in G-1 Plus media (Vitrolife) and were transferred to G-2 Plus media (Vitrolife) on day 3. Slides containing embryos were placed into the Embryoscope chamber immediately and cultured at 37.5 °C in 6% $CO_2$, 5% $O_2$, and 89% $N_2$. The morula- and blastocyst-stage embryos were collected at day 4 or day 6 after fertilization. All of the embryos used in this study have good morphology with appropriate developmental speed. The embryonic assessment was performed as described previously[35]. The embryos were transferred into Acidic Tyrode's Solution to remove the zona pellucida. Zona-free embryos were incubated for 20 min (for morula) or 30 min (for blastocyst) in Accutase medium before dissociating into single blastomeres by careful pipetting. Then washed thoroughly in PBS with 0.5% (m/v) BSA. Single blastomeres were isolated by gentle, repeated pipetting. The separated blastomeres washed 3–5 times in PBS with 0.5% BSA and placed into 200-μl PCR tube for scCAT-seq library preparation.

**Immunofluorescence staining**. The blastocyst embryos were first treated with acidic Tyrode's solution to remove the zona pellucida. After washing, the blastocysts were fixed with 4% paraformaldehyde (Sigma, #30525-89-4) for 30 min at the room temperature and washed three times in PBS supplemented with 0.1% BSA, and then subjected to membrane permeabilization with 1% Triton X-100 (Sigma, #T8787) for 30 min. After washing, the blastocysts were blocked in a blocking solution containing 5% donkey serum albumin (Jackson ImmunoResearch, #017-000-121) and 2% BSA in PBS. After blocking at 4 °C overnight, blastocysts were incubated with rabbit anti-NANOG (1:100; Abcam, #Ab109250) and goat anti-SOX17 (1:40; R&D, #AF1924) at 4 °C overnight. After washing five times, the samples were incubated with Alexa Fluor 488 donkey anti-rabbit IgG (1: 1000, Thermo, #A21206) or Alexa Fluor 594 donkey anti-goat IgG (1: 1000, Thermo, #A11058) for 1 h at 37 °C. DNA was stained (15 min incubation, 37.5 °C) with DAPI dye (1 μg/ml, Invitrogen, #D1306). Fluorescent cells were visualized and digital images were captured using the inverted confocal microscope.

**Single-cell CAT-seq**. The scCAT-seq protocol can be done manually or by conventional liquid-handling robots for parallel processing of multiple single cells (e.g., 96 cells in this study). Single cells were sorted by flow cytometry into a 96-well plate and lysed in a 7 μl mild lysis buffer (10 mM NaCl, 10 mM Tris-HCl, pH 7.5, 0.2% IGEPAL CA-630 (0.4% for single blastomeres), 10 U RNase-inhibitor (NEB)) for 15 min at 4 °C (note that the concentration of IGEPAL CA-630 could be optimized for different cell types). The lysate was vortexed for 1 min and was then centrifuged at 2000 g for 5 min in a refrigerated centrifuge to leave the nucleus at the bottom of the well. 4 μl of lysis product supernatant (containing the RNA content) was carefully transferred into another 96-well plate supplemented with 0.5 μl ERCC spike-in mixture (1: 250,000 dilution, Ambion), 1 μl of 10 mM dNTP mix (Enzymatics), and 1 μl of 10 μM modified oligo-dT primer (5′-AAGCAGTGGTA TCAACGACTACT30VN-3′, where V is either A, C, or G, and N is any base) and then incubate at 72 °C for 3 min.

Note that the physical separation procedure is critical for the successful capture of chromatin and RNA content. The single nucleus in the bottom of each well could be validated by qPCR using a two-step amplification strategy: (1) amplify the whole transposed DNA for eight cycles using primers targeting Tn5 adaptor (for: 5′-TCGTCGGCAGCGTCAGATGTGTATAAGAGACAG-3′, rev: 5′-GTCTCGTGGGCTCGGAGATGTGTATAAGAGACAG-3′); (2) amplify the DNA fragment within a generally accessible region PCR primers (for: 5′-GGTCTGAACTGTTGGGTGCT-3′, rev: 5′-GGGCTGTGAATTCAGGCTTA-3′).

Immediately after the separation step, 8.5 μl of a reverse-transcription master mix (150 U SuperScript II reverse transcriptase (Invitrogen), 15 U RNase-inhibitor, 1x SuperScript II First-Strand buffer, 0.75 μl of 0.1 M DTT, 3 μl of 5 M betaine (Sigma), 0.09 μl of 1 M $MgCl_2$ (Millipore), 0.15 μl of 100 μM Template-Switching Oligo (5′-AAGCAGTGGTATCAACGCAGAGTACATrGrG + G-3′, where "r" indicates a ribonucleic acid base and "+" indicates a locked nucleic acid base, Exiqon) and NF-water) was added to each well. The mixture was then thermal cycled as follows: 42 °C for 90 min, 10 cycles of 50 °C for 2 min, 42 °C for 2 min, and finally 70 °C for 15 min. Afterward the PCR master mix (15 μl KAPA HiFi HotStart ReadyMix with 0.3 μl of 10 μM PCR primer (5′-AAGCAGTGGTATCA ACGCAGAGT-3′)) was added to the reverse-transcription reaction mixture and thermal cycled as follows: 98 °C for 3 min, 18 cycles of 98 °C for 20 s, 67 °C for 20 s, 72 °C for 6 min, and finally 72 °C for 5 min. Amplified cDNA was purified using

KingFisher Flex purification instrument with using a 1: 1 volumetric ration of AMPure XP beads (Beckman Coulter) and eluted into 25 μl NF-water.

During the RNA library preparation process, the precipitated nuclei were resuspended in a 4 μl transposase reaction mix (1x TAG buffer, 0.3 μl Tn5 transposase (0.8 U/ul) and NF-water). The transposition reaction was carried out for 15 min at 37 °C. Then 3.5 μl mix of stop buffer (2.1 μl of 0.1 M EDTA, pH 8.0, 0.42 μl of 0.1 M Tris-HCl, pH 8.0, and NF-water) was added and the reaction was maintained at 50 °C for 15 min. To minimize the DNA loss and maximize the yield of the extremely small amount of transposed DNA (<0.1 pg) from single nucleus, we added a large amount of plasmid DNA (30 ng) as a carrier DNA together with 3 μl of RLT Plus buffer (QIAGEN) to the mixture immediately after the stop step. The lysis process was performed with shaking on a thermomixer for 15 min at 37 °C. Afterward, the DNA was purified using KingFisher Flex with a 1:1.8 volumetric ration of XP beads. Finally, the DNA was eluted with 25 μl NF-water. We used a 50 μl PCR amplification mix (transposed DNA, 25 μl NEBNext High-Fidelity 2x PCR Master Mix, 0.5 μl of 20 μM transposase adapter 1 (5′-TCGTCGGCAGCGTCAGATGTGTATAAGAGACAG-3′), 0.5 μl of 20 μM adapter 2 (5′-GTCTCGTGGGCTCGGAGATGTGTATAAGAGACAG-3′)) to amplify the DNA and then proceeded to perform eight cycles of PCR using the following conditions: 72 °C for 5 min; 98 °C for 1 min; and thermocycling at 98 °C for 15 s, 63 °C for 30 s, and 72 °C for 1 min. The pre-amplified transposed DNA was harvested using KingFisher Flex with a 1:1 volumetric ration of XP beads and finally eluted in a total of 25 μl NF-water.

For chromatin accessibility libraries, DNA was amplified for another 10–16 cycles (The number of cycle could be evaluated by qPCR analysis for different cell types[18], but based on our experience the appropriate number of cycles are 8 cycles for samples of 500 cells, 12 for samples of 10 cells, and 15 for samples of single cell) using the following PCR reaction mixture: pre-amplified transposed DNA, 25 μl NEBNext High-Fidelity 2x PCR Master Mix, 1 μl of 20 μM universal primer, 1 μl of 20 μM barcode primer. For sequencing, DNA were size-selected with XP beads for fragments between 150 and 700 bp in length according to the manufacturer's instruction, and finally eluted with 25 μl of TE buffer. For RNA libraries, 2 ng cDNA were used for the tagmentation reaction carried out with 10 μl mixture containing 0.3 μl transposase, 1x TAG buffer and NF-water. The tagmentation reaction was incubated at 55 °C for 10 min and released Tn5 with 2.5 μl of 0.1% SDS. The transposed cDNA was then used for PCR amplification and library preparation according to the Smart-seq2 method described previously[7].

All libraries were further prepared based on BGISEQ-500 sequencing platform[36]. In brief, the DNA concentration was determined by Qubit (Invitrogen). After that, 2 pmol pooled samples were used to make single-strand DNA circle (ssDNA circle). Then DNA nanoballs (DNBs) were generated with the ssDNA circle by rolling circle replication to enlarge the fluorescent signals at the sequencing process as previously described[36]. The DNBs were loaded into the patterned nanoarrays and sequenced on the BGISEQ-500 sequencing platform with pair end 50-bp read length.

**Transcriptome data processing**. The raw reads of transcriptome data were firstly aligned to Human rRNA sequence including 28S (NR_003287.2), 18S (NR_003286.2), 5S (NR_023379.1), and 5.8S (NR_003285.2) using SOAP2[37]. The mapped reads were filtered using custom script. The retained reads were mapped to hg19 genome using HISAT[38] with the parameters: --sensitive --no-discordant --no-mixed –I 1 –X 1000. Reads with mapping quality less than 30, and duplicate reads were discarded using samtools. The number of read within each gene in each single cell (GENCODE, v19) were counted using GenomicAlignments package[39] with parameters below: mode = "Union", inter.feature = TRUE and singleEnd = FALSE. The count matrices were supplied as supplementary data files (Supplementary Data 4 and 6).

**Chromatin accessibility data processing**. The raw reads of chromatin accessibility data were trimmed by custom script and aligned using Bowtie[40] (parameter: -X 2000 -m 1). Reads with mapping quality less than 30, and reads mapped to the mitochondria genome or the hg19 consensus excludable region (http://hgdownload.cse.ucsc.edu/goldenPath/hg19/encodeDCC/wgEncodeMapability/) were filtered out. Duplicate reads were removed using Picard *MarkDuplicates* function (http://broadinstitute.github.io/picard/). To obtain a unique peak list of all cell lines, we first adopt the model-based analysis of ChIP-seq (MACS2)[41] to call peaks using bam files from each bulk ATAC-seq profiles with the following parameters: --nomodel --nolambda --keep-dup all --call-summits. Afterward, the peaks from different cell lines were merged as a unique peak list. For human embryos, bam file merged from those of all usable single cells was used for peak calling. The number of raw fragment within each peak in each single cell were counted using ChromVAR[29]. Peaks that were detected (with number of fragment more than 1) in less than 10% single cells were filtered out. The count matrices were supplied as supplementary data files (Supplementary Data 3 and 5).

**Calculating the single-cell chromatin accessibility fragment density in different genomic contexts**. The peak regions of ChIP-seq profiles for histone modifications and transcription factors (TFs) in this study were downloaded from ENCODE (Supplementary Data 2). The chromatin accessibility peaks overlapping each ChIP-seq region were determined by bedtools[42] *intersection* function. Genes

located 5 kb upstream or downstream each peak are assigned as putative target genes of the peak. Genes were defined as active, bivalent, inactive gene classes based on the enrichment of H3K4me3, H3K27ac, and H3K27me3 at their regulatory regions:[20] (1) active genes, which show the co-enrichment of H3K4me3 or H3K4me1 and H3K27ac, and the absence of H3K27me3; (2) bivalent genes, which show the co-enrichment of H3K4me3 or H3K4me1, and H3K27me3; (3) inactive genes, which show the enrichment of H3K27me3, but the absence of H3K4me3, H3K4me1 and H3K27ac. The fragment density was determined by computing CPM (counts per million) values of each peak at each single cell.

**Inferring regulatory links between genomic features**. Regulatory links between chromatin accessible regions and target genes were identified based on scATAC-seq data and scCAT-seq data. Only expressed genes and accessible peaks in more than 10% of the cells were used and normalized by deconvolving size factors from cell pools[43]. For scATAC-seq data, we assigned regulatory links based on the correlation between the signal of distal peaks and peaks in the promoter. For scCAT-seq data, we used the correlation between the signal of distal peaks and the target gene expression. To avoid underestimating the computed correlation as a consequence of intrinsic differences between cell subpopulations, we computed a weighted Spearman correlation using the R package wCorr[44]. A weighted Spearman correlation was computed for each NMF signature, using the corresponding exposure to the NMF H matrix as weights, and a regulatory link was assigned if at least one of the computed correlation was greater than 0.25.

**Inferring scCAT-seq based regulatory relationships**. Single-cell-specific regulatory relationships between genes and their nearby accessible regions (1 Mb upstream-downstream) were assigned using the scCAT-seq data following a three steps strategy: (1) identification of active TFs for every cell by pySCENIC[22], using the normalized gene expression matrix: regulons were defined based on the co-expression of TFs and their target genes across cells. Regulon enrichment was characterized in each cell by measuring the area under the recovery curve (AUC) of the genes that defined each regulon. Finally, individual TFs were defined as active or inactive in each cell based on the bimodal distribution of the AUC scores of the corresponding regulon. (2) Identification of active, accessible regions: The binding motifs of active TFs were matched to accessible regions using the Biostrings R package[45]. Accessible regions were labeled as active for each cell when at least one motif matched with at least 95% of the highest possible score for the given motif Position Weight Matrix (PWM). (3) Regulatory relationships assignment: a Wilcoxon test was applied for each gene to determine if the presence of a nearby active, accessible region was associated with a significative change in its expression. All regions around 1 Mb of each gene were tested to assign a regulatory relationship between them when the resulting p-value was less than 0.05. Accordingly, each gene could have more than one regulatory relationship, reflecting the complexity of the cell regulatory landscape. Finally, to recover genomic signatures based on the regulatory patterns shared between cells, NMF was applied to the binary matrix of regulatory relationships using the R package Bratwurst[26].

**NMF clustering analysis**. We used a new implementation of the NMF algorithm in the R package Bratwurst[26], in order to decompose each matrix into a exposure matrix H and a signatures matrix W, for factorization ranks $K \in \mathbb{Z} : K \in [2,6]$. The optimal factorization rank was selected as the K that best satisfies the quality metrics criteria: minimize the Frobenius error and the mean Amari distance, while maximizing the cophenetic correlation coefficient. Subsequently, K signatures were identified from the H matrix, and specific features were identified for each signature after performing feature extraction from the W matrix. These specific features contribute exclusively to one single signature. To evaluate the similarity between signatures at different factorization ranks, the normalized non-negative linear least squares estimates were computed across all factorization ranks' W matrices to the next factorization rank W matrix, with the Bratwurst package[26].

**Computing the chromatin accessibility deviations for TFs**. The R package ChromVAR[29] was applied to compute the chromatin accessibility deviation scores. The candidate TF motifs are from MotifDB database[46]. The variability of each TF was computed by the *computeVariability* function. The deviation score of each TF was computed using the *computeDeviations* function.

**Reporting Summary**. Further information on experimental design is available in the Nature Research Reporting Summary linked to this article.

## Data availability

All raw data were deposited in the Sequence Read Archive (SRA) of NCBI (accession code: SRP167062). These data were also deposited in the CNGB Nucleotide Sequence Archive (accession code: CNP0000213). All other relevant data are available upon request.

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

## Acknowledgements

We thank all members of the Stem Cell and Development Lab (BGI) for their support and Scott Edmunds, Christian Conrad, Kun Ma and Ying Shan for helpful discussion. We also thank Jijun Cheng, Yuan Long, and Feifei Zhang from Shanghai LIDE Biotech Co., Ltd. for technical support. This work was supported by the Strategic Priority Research Program of the Chinese Academy of Sciences, Grant No. XDA16010402, the Shenzhen Municipal Government of China Peacock Plan (KQTD20150330171505310), and the Shenzhen Engineering Laboratory for Innovative Molecular Diagnostics (DRC-SZ (2016) 884). Longqi Liu is funded by the China Postdoctoral Science Foundation (2017M610553). Andrés Quintero is funded through a NCT3.0/DKFZ grant within the ENHANCE project.

## Author contributions

L. Liu, C.L., Z.S., and X.X. conceived the idea. L. Liu, C.L., and Y. Yuan designed the scCAT-seq method. C.L. and Y. Yuan performed the majority of the experiments and generated the scCAT-seq data. L.W. and L. Liu performed preprocessing and quality evaluation of all scCAT-seq data. A.Q. and C.H. developed the algorithms for regulatory relationship inferring and NMF clustering. A.Q. and C.H. performed regulatory relationship analyses for all datasets of the cell lines. L. Liu performed the integrative analyses for the datasets of human embryos. L. Leng and G.L. collected the embryo samples and performed the immunostaining experiments. M.W., M.C., L.X., G.D., R.L., J.X., X.C., H. L., Q.Z., X.L., G.L., and Quanlei Wang assisted with the experiments. Qi Wang, D.C., Y. L., S.L., and X.W. assisted with the data analyses. L. Liu wrote the paper with input from C.L., L.W., A.Q., and Y. Yuan. L. Liu, C.L., L.W., A.Q., and Y. Yuan prepared the figures. Z.S., C.H., A.Q., L.F., R.E., Z.G., and X.X. revised the paper. H.W., X.L., H.Y., S.Z., Y.H., Y. Ye, and F.C. provided helpful comments on the manuscript. X.X., L. Liu, and Z.S. supervised the entire study, R.E. supervised the algorithm development. All authors read and approved the manuscript for submission.
