## [Peer Review File · Nature Communications]

REVIEWERS' COMMENTS:

Reviewer #2 (Remarks to the Author):

The current version of the paper is improving some aspects of the analysis, and the paper is more focused than before, presenting the methods and then some applications around it. I still do not think the computational methodology is appropriately supporting all conclusions on regulatory relationship, but given that this is not the main result in the paper and that some of the data will be useful for further development in the field, I support publication of this work.

I would urge the authors to make data fully available, including tables of all inferred RNA levels and accessible sites per cell, possibly as a supplementary table to the paper or as table supporting their submission of the raw data.

I also urge the authors to release code generating the main figure panels from such tables. In particular, precise parameters and improved description in the method section "Inferring scCAT-seq based regulatory relationships" should be enhanced with code implementing the strategy. In its current form it is far from clear and there are minimal validations or tests as to the specificity of the approach.

In addition, I believe most of the concerns raised by reviewer 1 are addressed, primarily by removing elements of the manuscript that were not well supported. Some concerns regarding methodology (MMF) were only partially addressed, but I believe this is easy to address by fully describing the analytic pipeline (through releasing of code and supplementary tables).

Reviewer #3 (Remarks to the Author):

The manuscript has been significantly improved. The authors have made a substantial effort to rephrase and clarify parts that are retained and they have removed the problematic analyses. Additionally, the authors have added data from pre-implantation embryos and introduced more analyses strategies and results for combined ATAC-seq and RNA-seq data on the single-cell level. Overall, the authors have addressed my concerns satisfactorily and I am now supporting publication.

Reviewers' comments:

Reviewer #2 (Remarks to the Author):

Item 1: The current version of the paper is improving some aspects of the analysis, and the paper is more focused than before, presenting the methods and then some applications around it. I still do not think the computational methodology is appropriately supporting all conclusions on regulatory relationship, but given that this is not the main result in the paper and that some of the data will be useful for further development in the field, I support publication of this work.

I would urge the authors to make data fully available, including tables of all inferred RNA levels and accessible sites per cell, possibly as a supplementary table to the paper or as table supporting their submission of the raw data.

Answer: All the raw scCAT-seq datasets were deposited to NCBI Sequence Read Archive (accession code: SRP167062) and CNGB Nucleotide Sequence Archive (accession code: CNP0000213). The processed read count matrices indicating the RNA levels and accessible sites per cell were provided as Supplementary Data files (Supplementary Data 3-6).

Item 2: I also urge the authors to release code generating the main figure panels from such tables. In particular, precise parameters and improved description in the method section "Inferring scCAT-seq based regulatory relationships" should be enhanced with code implementing the strategy. In its current for it is far from clear and there minimal validations or tests as to the specificity of the approach.

Answer: Computer codes used for processing and evaluation of scCAT-seq data are available at <https://github.com/single-cell-BGI/scCAT>. The codes used for inferring regulatory relationships are available at: <https://github.com/hdsu-bioquant/scCAT>.

Item 3: In addition, I believe most of the concerns raised by reviewer 1 are addressed, primarily by removing elements of the manuscript that were not well supported. Some concerns regarding methodology (MMF) were only partially addressed, but I believe this is easy to address by fully describing the analytic pipeline (through releasing of code and supplementary tables).

Answer: We have now released the codes and data matrices as described above. We thank the reviewer for supporting the publication of our study.

Reviewer #3 (Remarks to the Author):

The manuscript has been significantly improved. The authors have made a substantial effort to re-phrase and clarify parts that are retained and they have removed the problematic analyses. Additionally, the authors have added data from pre-implantation embryos and introduced more analyses strategies and results for combined ATAC-seq and RNA-seq data on the single-cell level. Overall, the authors have addressed my concerns satisfactorily and I am now supporting publication.

Answer: We thank the reviewer for supporting the publication of our study.